# How E-learning Is Correlated with Competitiveness and Innovation and Critical Success Factors

**Gilnei Alberto Lopes [1], João Carlos Furtado [1] and Ismael Cristofer Baierle [2,\***

[1] Industrial Systems and Processes, Universidade de Santa Cruz do Sul-UNISC, Santa Cruz do Sul 96815-900, Brazil; gilnei@polouinter.com.br (G.A.L.); jcarlosf@unisc.br (J.C.F.)
[2] School of Engineering, Universidade Federal do Rio Grande-FURG, Rio Grande 96203-900, Brazil
\* Correspondence: ismaelbaierle@hotmail.com; Tel.: +55-51999956774

**Abstract:** E-learning has gained a prominent role in the education scenario, either because of its capacity for extraterritorial coverage, or because of the scale it offers for free and academic courses. How e-learning is being managed and identifying opportunities for improvement in this process is a challenge for managers. A systematic review of the literature on e-learning was carried out from the perspective of process management. The Scimat and VOSviewer software were used together to make it possible to understand the volume of publications, terms, density, and perspectives of studies on the subject on the indexing platforms, as well as pointing out challenges and trends in the area. The term e-learning does not appear as a trend driver in published articles. When related to the terms critical success factors (CSFs) and competitiveness and innovation, the greatest concentration of articles is directed to the e-learning infrastructure or technologies applied to it. As a result, it is possible to observe that the co-occurrence of e-learning with critical success factors and competitiveness and innovation is in its early stages, with scarce research in this area indicating room for future growth. E-learning entails unique business metrics that require specific tools and knowledge of technology, concepts, and the organizational environment. However, there is a dearth of publications addressing these aspects and proposing relevant methodologies and processes.

**Keywords:** e-learning; critical success factors; competitiveness; innovation; VOSviewer; Scimat

## 1. Introduction

E-learning is transforming higher education through the use of disruptive technologies [1]. This teaching modality has become increasingly important in recent decades [2]. It has been a differentiating tool for many universities because it focuses on the student, has interactive methods based on technology, and allows students to access information more easily [3]. E-learning allows synchronous or asynchronous learning, anywhere, just by having access to the internet [4]. The evolution of e-learning took place in different ways in different countries around the world. It began in the United States in 1988 with Power On! through the US Congressional Office of Technology Assessment. It continued in the European Union, through the plan "Europe and the Global Information Society" in 1994. Moreover, there was the beginning of e-learning policies in Australia in the 1990s. This trend started in South Korea as well in the 1990s [5]. The traditional teaching model underwent a migration to new approaches with e-learning tools [6], providing greater access to knowledge to more people.

The innovation process results in improvements mainly within organizations. The innovation process refers to the systematic approach of generating and implementing new ideas, technologies, products, or processes that result in significant improvements or breakthroughs. It involves various stages, including ideation, research, development, testing, implementation, and commercialization.

These improvements may include machinery, technology, and processes aiming to reduce costs and improve flexibility, efficiency, and competitiveness [7,8]. Strategic management is a

process of multiple paths, which can be grouped and categorized for better implementation to solve problems and generate results [9] that meet the organization's objectives.

*E-Learning*

E-learning has had impressive growth with the advent of technological advances, specifically in the area of education, from the integration of the tripod: technology, education, and economics [10]. This teaching modality presents, among other advantages, low cost, allocation of time to study, self-managed learning by the student, and flexibility of study places as long as there are consistent and standardized platforms [11].

As a social contribution, e-learning is changing the way people transfer knowledge, as it is no longer necessary to be physically present or on the same time schedule to access the class of the day. Studies on the subject have been carried out covering different technologies, methodologies, uses, and approaches in e-learning instruction [12]. In 2015, the global e-learning market was estimated at 165 billion USD with an average annual growth prospect of 5% from 2018 to 2023 [3]. For [13], being part of innovation environments and incorporating innovation into management processes is emerging as a global trend. Therefore, companies operating in this sector must understand how e-learning is incorporating management tools to increase e-learning management under the prism of increased performance and managerial qualification [14].

To achieve their strategic goals, some organizations use a set of key ideas that must be carefully considered and followed: the so-called critical success factors (CSFs) [15]. In the context of e-learning, critical success factors (CSFs) refer to the key factors that are essential for the successful implementation and outcomes of e-learning initiatives or programs. These factors influence the effectiveness, efficiency, and overall success of e-learning efforts. Some common CSFs in e-learning: (a) Clear Learning Objectives: Clearly defined and measurable learning objectives are crucial for e-learning success. They provide a clear focus and direction for designing and delivering the online courses or programs; (b) engaging and Interactive Content: e-learning content should be engaging, interactive, and designed to promote active learning. This includes multimedia elements, such as videos, interactive exercises, simulations, and assessments, to enhance learner engagement and knowledge retention; (c) User-Friendly Learning Platform: the e-learning platform should be user-friendly, intuitive, and accessible. It should provide a seamless learning experience for learners, allowing them to navigate easily, access resources, participate in discussions, and track their progress; (d) Adequate Technical Infrastructure: a reliable and robust technical infrastructure is essential for smooth e-learning delivery. This includes stable internet connectivity, compatible devices, and appropriate software or learning management systems (LMSs) that can support the required functionalities; (e) Learner Support and Engagement: providing adequate learner support is crucial in e-learning. This can include access to instructors or mentors, discussion forums, online help resources, and timely feedback to address learners' questions and concerns. Engaging learners through regular communication, feedback, and collaborative activities is also important; (f) Effective Assessment and Feedback: a well-designed assessment strategy that aligns with the learning objectives is necessary. It should include various types of assessments, such as quizzes, assignments, and projects, to evaluate learners' understanding and progress. Timely and constructive feedback helps learners improve and stay motivated; (g) Pedagogical Design and Instructional Strategies: applying effective pedagogical principles and instructional strategies in e-learning is critical. This involves designing instructional materials that accommodate diverse learning styles, providing opportunities for active learning and reflection, and incorporating real-world applications; (h) Continuous Monitoring and Evaluation: ongoing monitoring and evaluation of e-learning initiatives are essential to ensure their effectiveness and identify areas for improvement. This includes collecting and analyzing data on learner performance, feedback, and satisfaction, as well as making data-driven decisions to enhance the e-learning experience; (i) Scalability and Sustainability: e-learning initiatives should be designed with scalability and sustainability in mind. This involves considering

factors such as the potential for growth, the ability to update and adapt content, and the availability of resources to support ongoing e-learning activities.

These CSFs are not exhaustive and may vary depending on the specific context and goals of the e-learning program. It is important for organizations or institutions implementing e-learning to identify and prioritize the critical success factors that align with their unique requirements and desired outcomes.

The critical success factors (CSFs) must be analyzed very carefully for the organizational success of e-learning [16] with the help of intelligent systems to fulfill the role of a strategic management tool [17,18]. In this sense, it is worth analyzing the relationship of e-learning with innovation and competitiveness in scientific production.

Competitiveness in the context of e-learning refers to the ability of an e-learning program, platform, or institution to effectively position itself and stand out in the increasingly competitive online education market. It involves offering distinctive features, high-quality content, and a superior learning experience that attracts and retains learners in a highly competitive digital learning environment. Some key aspects of competitiveness in e-learning include: (a) Differentiation: to be competitive, e-learning providers need to differentiate themselves from others in the market. This can be achieved through unique course offerings, specialized programs, innovative teaching methods, or by catering to specific target audiences; (b) Quality Content and Instruction: offering high-quality instructional content is essential to remain competitive. This includes well-designed and up-to-date courses, interactive multimedia elements, engaging learning materials, and experienced instructors who can deliver effective online instruction; (c) Technological Infrastructure: competitiveness in e-learning relies on having a robust and user-friendly technological infrastructure. This includes reliable learning management systems (LMSs), seamless online platforms, responsive websites or applications, and easy access to course materials; (d) Flexibility and Customization: e-learning programs that offer flexibility in terms of scheduling, pacing, and learning pathways are attractive to learners. Customization options, such as personalized learning plans, adaptive learning technologies, and individualized support, can further enhance competitiveness; (e) Recognition and Accreditation: accreditation or recognition by relevant educational bodies or industry associations can enhance the competitiveness of an e-learning program. This adds credibility and demonstrates that the program meets certain quality standards and recognized benchmarks; (f) Affordability and Value for Money: offering competitive pricing structures and value for money is crucial in attracting and retaining learners. Providing transparent pricing, flexible payment options, and demonstrating the return on investment in terms of skills development or career advancement can contribute to competitiveness; (g) Continuous Improvement and Innovation: staying competitive in e-learning requires a commitment to continuous improvement and innovation. This involves monitoring industry trends, adopting emerging technologies, incorporating feedback from learners, and regularly updating course content to remain current and relevant; (h) Market Awareness and Marketing Strategies: understanding the target market, identifying learner needs, and developing effective marketing strategies are essential for competitiveness. This includes market research, targeted advertising, social media presence, and effective communication to attract and engage potential learners.

Competitiveness in e-learning is a dynamic process that requires a proactive approach to adapt to changing market demands, technological advancements, and learner preferences. By focusing on differentiation, quality, engagement, support, flexibility, and continuous improvement, e-learning providers can position themselves competitively and succeed in the online education landscape.

In a study published by [19] a review was carried out covering, among other aspects, the models, advantages, disadvantages, challenges, and critical success factors of e-learning. As for the latter, they were further categorized, demonstrating that some factors are more relevant than others from the perspective of interested groups in the area of technology and design. In another study, Alqahtani and Rajkhan [16] evaluate the critical success factors (CSFs) using the analytical hierarchy process (AHP) and the Order Preference Technique

(TOPSIS) seeking to understand which are the most relevant CSFs for e-learning in the face of the COVID-19 pandemic in 69 educational institutions. CSFs are relevant and their prioritization enables better organizational performance [17–19]. Identifying and prioritizing CSFs to improve performance in e-learning management is gaining ground as an organizational differential.

The objective of this study is to identify publications that deal with e-learning management with a focus on CFSs and competitiveness and innovation indicators; analyze the relationship of e-learning with CSFs and competitiveness and innovation, crossing this information in a structured way and identifying in the bibliography, if any, solutions already presented to this question; and evaluate the possibility of adapting methods to other business models similar to the studied business. To better understand these questions, the following research questions are presented: (i) Is the term e-learning related to the research of innovation and business management? (ii) What is the volume of scientific production covering the themes e-learning, critical success factors, and competitiveness and innovation? (iii) What is the trend in e-learning research as related to managing critical success factors?

Understanding how these terms are correlated makes it possible to understand the current status and how these management techniques are used, particularly as related to e-learning.

## 2. Material and Methods

To develop this study, a Systematic Literature Review (SLR) was carried out, through the research of articles in international databases, reading and evaluating the relevance or lack of relevance of the study, and conducting a bibliometric analysis of the chosen terms. Digital databases are designed to facilitate access when searching for information. These allow for speed and updated information in the search process of the research [20]. The development of this article is structured into 8 steps, as shown in Figure 1.

The first stage presents a study on e-learning and the characteristics of e-learning, the social, economic, and academic contributions of the study, as well as research questions. The second step contains the method used, the definition of the keywords for this study, and the software settings. Next, a bibliometric study is carried out, analysis of the three key terms through VOSviewer and Scimat software as the starting point of this study. The analyses and clusters created by the software systems follow the rules described in [21].

The next step presents the correlation of the term e-learning with the terms critical success factors and competitiveness and innovation. The sixth step addresses the research questions. The next two steps list suggestions for further studies on the topic and the bibliographies used in this study, respectively, seeking to meet the proposal in this study.

In this study, a systematic literature review was performed according to [22]. The articles searched can be found in the international databases, Scopus, Web of Science, and Science Direct. For [13], the databases facilitate access to information due to their speed and updating capabilities. Articles were researched aiming to identify the main authors, the evolution of scientific production in this area over time, geographic regions that publish the most on the topic, which is the most significant keywords, and relevance and correlation between the themes of this study.

To understand the correlation between the studied themes, the VOSviewer [23] and Scimat [24] software were used in addition to the bibliometric data available in the Scopus platform. As they say [25] in the hypothesis, quality indicators have different views of quality, these can be complementary to form a new concept of quality. Assuming this line of thought, we have the combination of the three analytical methods which form a deeper and more comprehensive concept on the subject.

The terms chosen seek to improve our understanding of e-learning and its relationship with CSFs and competitiveness and innovation. The search terms defined were: "e-learning", "competitiveness and innovation"; and "critical success factors", following the criteria in Table 1.

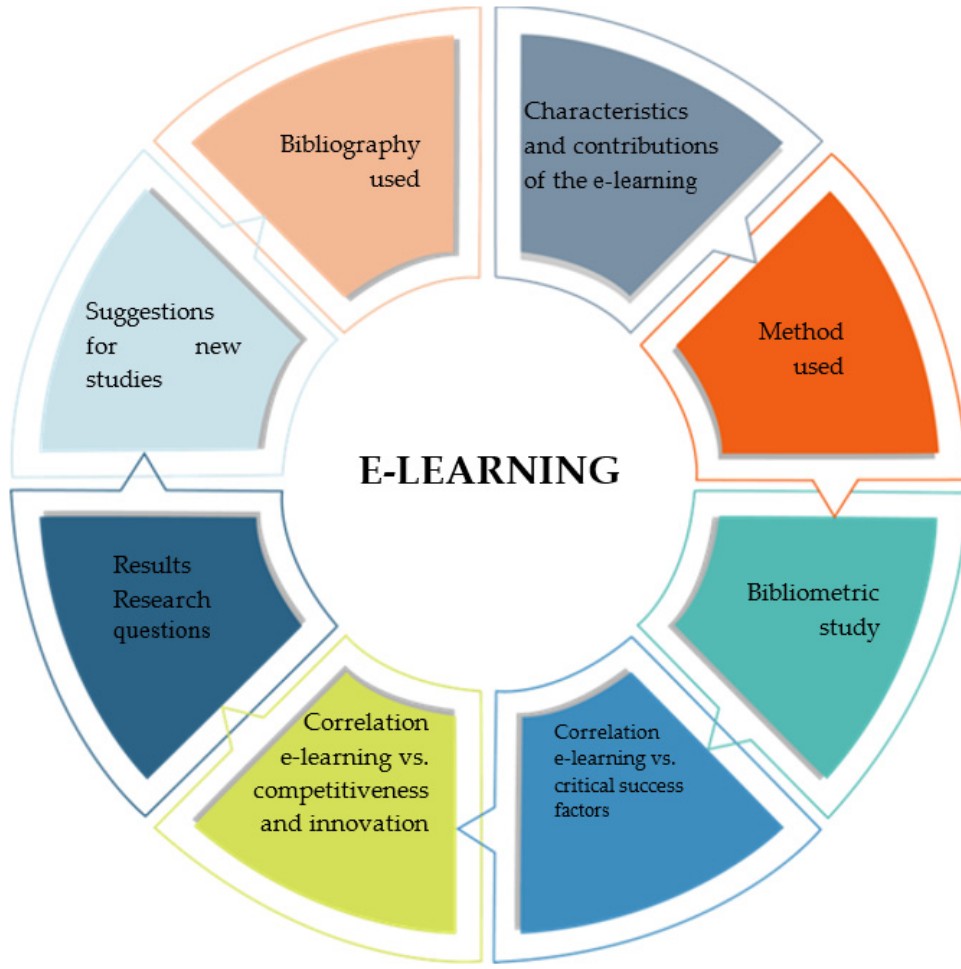

**Figure 1.** Article steps.

**Table 1.** Research criteria.

| Criteria | Database | | |
|---|---|---|---|
| | **Scopus** | **Science Direct** | **Web of Science** |
| Search Terms | "e-learning"; "competitiveness and innovation"; "critical success factors" | "e-learning"; "competitiveness and innovation"; "critical success factors" | "e-learning"; "competitiveness and innovation"; "critical success factors" |
| Studied area | Business, Management and Accounting, Decision Sciences | Business, Management and Accounting, Decision Sciences | Education Educational, Research Management, Business |
| Type of publication | Article and revision | Article and revision | Article and revision |
| Classification | Highest number of citations. | Greater relevance | Greater relevance |

Source: developed by the author.

The research design identifies how the study is structured. It explains how data are collected and defines sample criteria, methods used, and triangulations [26,27]. Figure 2 shows the research flow.

The steps shown in Figure 2 were important for the replication of this study to be as easy as possible. A brief explanation of each step is presented below for a better understanding of the construction of this study:

- Definition of the research topic: moment of reflection on the idea of the research having or not relevance as a contributor of knowledge. Being significant and relevant to the scientific production on the subject of e-learning;
- Formulation of the research problem: once the potential of the research topic was identified, this was the moment when the questions about the topic emerged. This step was the definition of which questions the study would propose to answer;
- Delimitation of the criteria: with the possession of the questions to be answered in this study, it was defined which search criteria would be used in the databases. The objective was to follow equal search parameters for published scientific productions;
- Data collection method: definition on which databases would be searched, which software would be used for the treatment of the collected information and criteria for setup of these software;
- Data interpretation: after collecting and treating the data, this was the moment when the results were evaluated according to the research questions defined in the problem formulation stage;
- Conclusion of the study: phase in which the data and interpretations are presented and compared with the existing literature in order to answer the research questions as well as contribute to possible new studies.

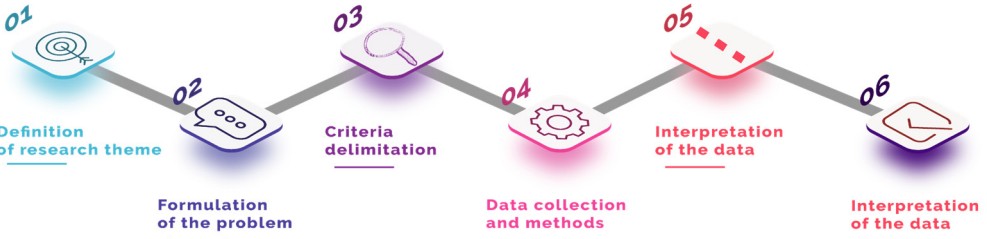

**Figure 2.** Research flow in the first stage of the study.

*2.1. Definition of Keywords*

To define the keywords for this research, the following terms were used:

- (A) "competitiveness" and "innovation";
- (B) "e-learning"; and
- (C) "critical success factors".

After the results, the filters "Article" and "Review" were applied, within the search result of the previous term. The results of the volume of articles found are shown in Figure 3.

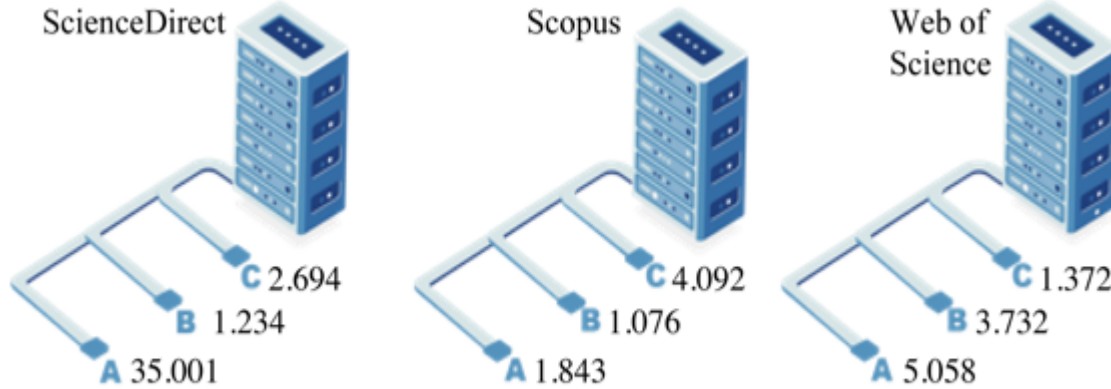

**Figure 3.** Volume of articles found.

### 2.2. Keyword Evaluation Methods

As a way to analyze the data collected on the digital platforms, the analysis of the data results provided by the Scopus platform was used in conjunction with the VOSviewer and Scimat software. VOSviewer software 1.6.16 allows for studies on word grouping or word grouping in similarity clusters, thus enabling the understanding of how researchers have directed their studies by areas in time. The parameters used for making the maps were: when choosing the type of data, the map was based on bibliographic data; the data source was data from bibliographic database files; the type of analysis and counting method was co-occurrence with the full count method; and the unit of analysis selected all keywords.

Scimat represents the correlation of words through a strategic diagram, thematic network structure, and the structure of the evolution of the [27–29]. Moreover, according to the authors, the software uses a clustering algorithm to detect themes through clusters graphically represented in two-dimensional form divided into four quadrants, demonstrating the density (y-axis) that measures the strength of the relationship between the keywords of a theme, and centrality (x-axis), which measures the intensity of how each cluster relates to other clusters.

The classification of themes follows a two-dimensional map: (a) motor themes, with high centrality and density, which are the most important because they have a high degree of development for research; (b) Basic and Transversal Themes: high centrality and low development, which present the possibility of becoming engines in the future; (c) Emerging or declining themes: low centrality and density, they need quality analysis to define whether they are emerging or declining; (d) Highly Developed and Isolated Themes: low centrality and high development, loss of importance due to the emergence of something new, a concept or a technology [24]. Figure 4 shows the strategic graph and its division by quadrants.

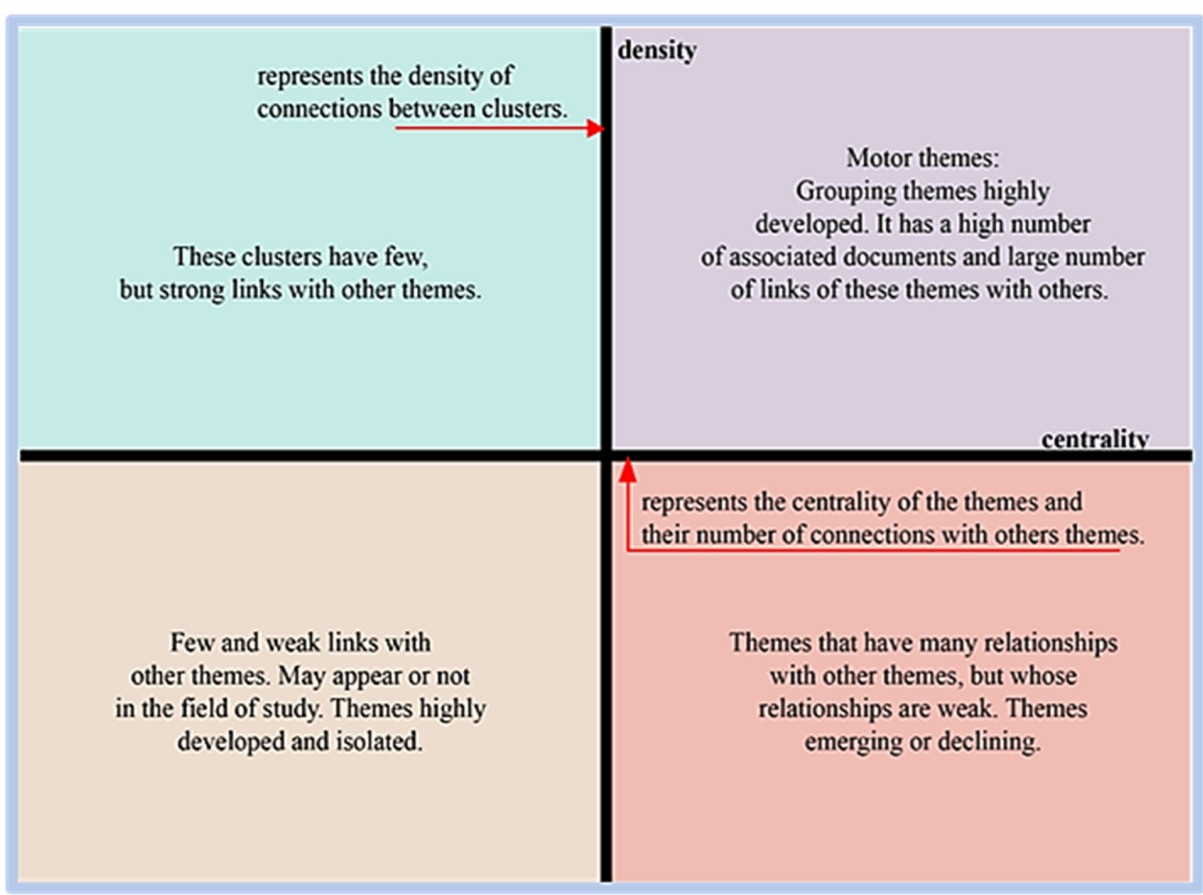

**Figure 4.** Scimat strategy chart.

In the structure of the thematic network, each cluster represents the words associated with this theme. The larger the cluster is, the greater the number of associated words, while the intensity of the correlation between the clusters is represented by the lines that interconnect them. The thicker the lines are, the stronger the correlations.

To equalize the analysis of the researched terms, four periods of analysis were defined. This segmentation allows for fairness in the analysis and a greater possibility of viewing the evolution of themes over time. The parameters used to configure the software are shown below:

- Definition of periods studied (by year):
  - ⇒ Until 1990;
  - ⇒ from 1991 to 2000;
  - ⇒ from 2001 to 2010;
  - ⇒ from 2011 to 2021.

### 3. Results

*3.1. E-Learning*

The scientific production using the term e-learning had an advance from 1992, with its peak of publications in 2021 with 3669 documents found. The author who has published the most on this topic is Hwang, Gwojen, National Taiwan University of Science and Technology, Taipei, Taiwan.

The VOSviewer analysis of the clusters formed from the term "e-learning" demonstrates a strong concentration of word density. In the visual representation for the term, there is a strong correlation with the words education, learning, internet, teaching, and teaching systems. However, there is less of a correlation with the words training, innovation, web 2.0, online education, design, customer satisfaction, among others. Figure 5 presents the word correlations in more detail.

When relating the term "e-learning" to the terms "critical success factors", and "competitiveness and innovation", the correlation is weak with little relevance. When relating the term "e-learning" to the term "critical success factors", it is noted that the correlation between the terms is weak with little relevance. Figures 6 and 7 show this correlation.

When comparing the term "e-learning" with the term "innovation", we can see greater relevance and proximity of clusters, as shown in Figure 8:

Using Scimat, we can verify the evolution of words associated with the term e-learning over time. Figure 9 shows this evolution:

We can observe that in the first period, the use of the term is practically insignificant. Over time, the use of the word e-learning gains space in scientific production, reaching its peak in the fourth period studied, from 2011 to May 2021, with a cluster of 6471 word representations.

As for the density and centrality of words in the previously defined periods, we have:

- Period 01—until 1990: no graphical representation;
- Period 02—from 1991 to 2000: no graphical representation;
- Period 03—from 2001 to 2010: graphical representation in Figure 10.

We can see that in this period, the word e-learning has no significant representation, something close to the theme as "learning organizations" is identified with low density and centrality.

When evaluating the correlation of the e-learning cluster we observed the strongest links between "computer-based internet" and "education". When we evaluate period 04, however, there is a significant change as shown in Figure 11 below:

This period presents a significant variation from the previous period. We observe "e-learning platform" as a search engine theme, but with a small cluster. The strongest correlation is the theme "learning systems" followed by "education".

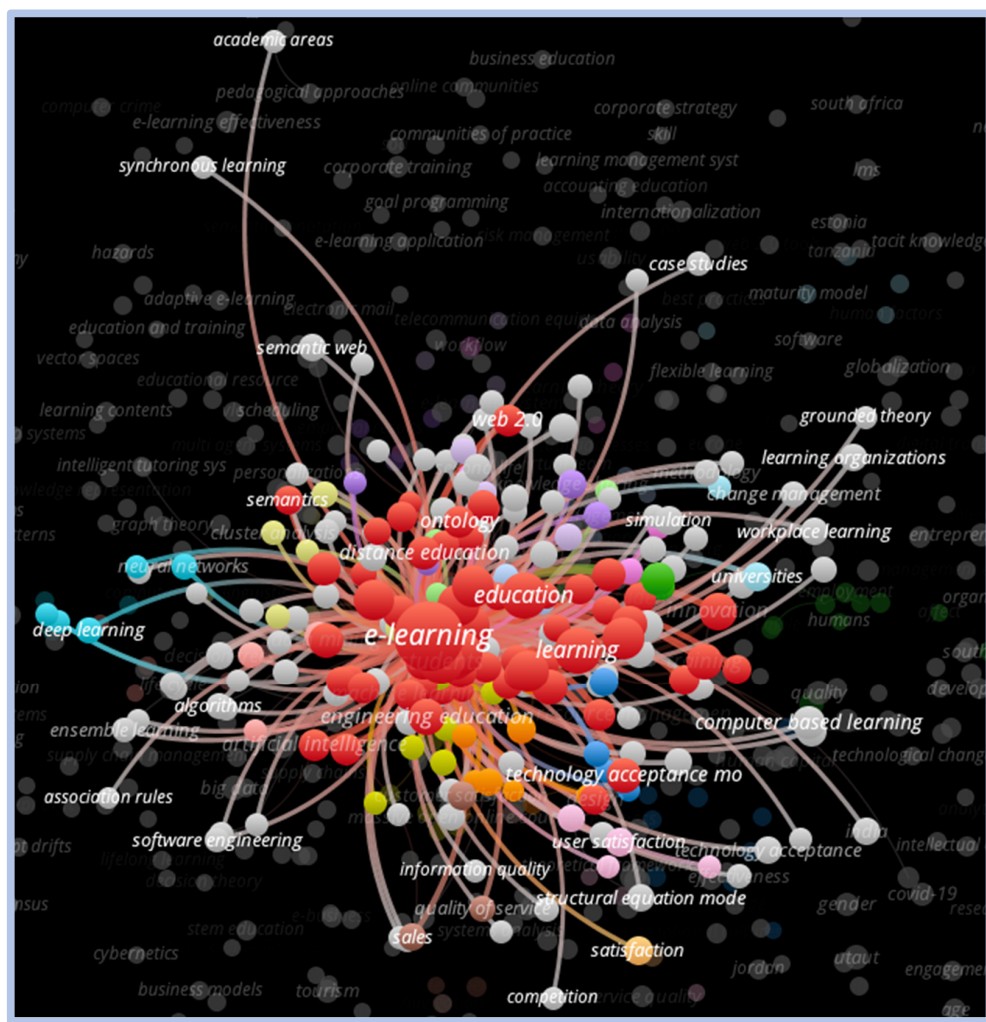

**Figure 5.** Network visualization—e-learning.

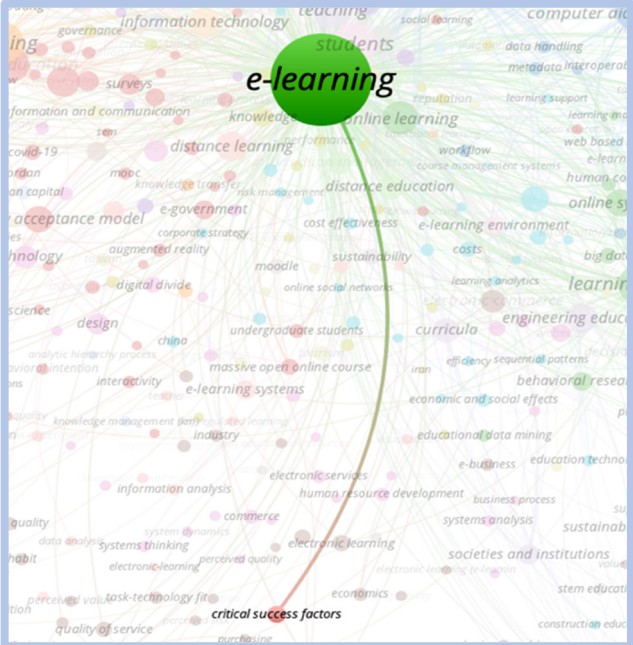

**Figure 6.** Correlation e-learning vs. critical success factors.

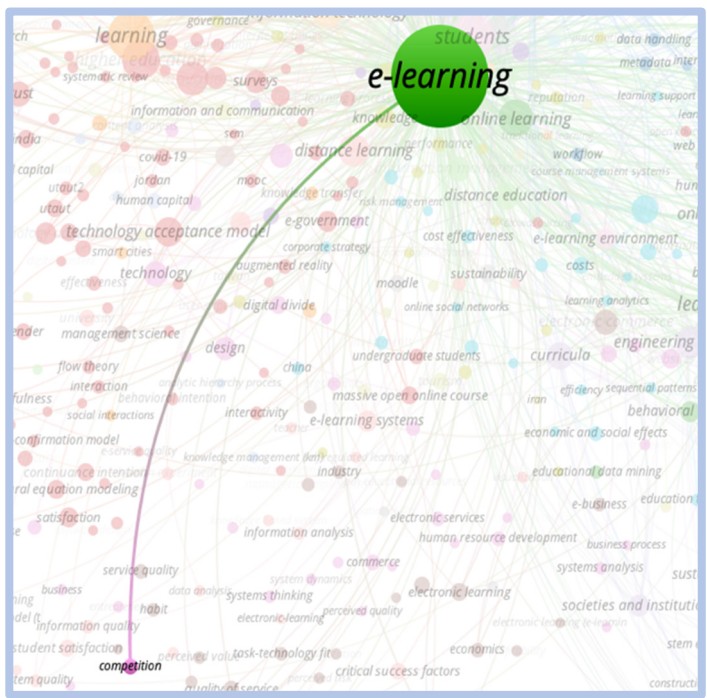

**Figure 7.** Correlation e-learning vs. critical success factors.

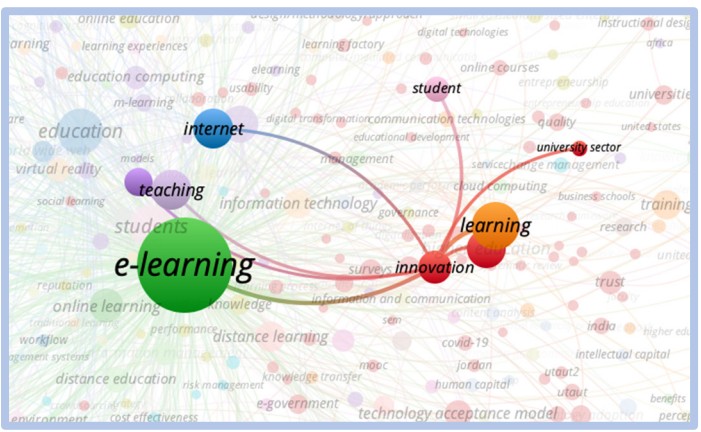

**Figure 8.** Correlation—E-learning vs. innovation.

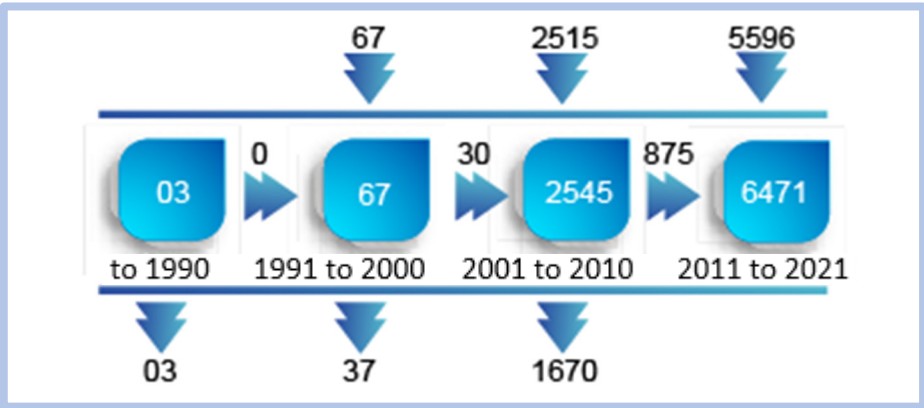

**Figure 9.** Evolution of words over time—e-learning.

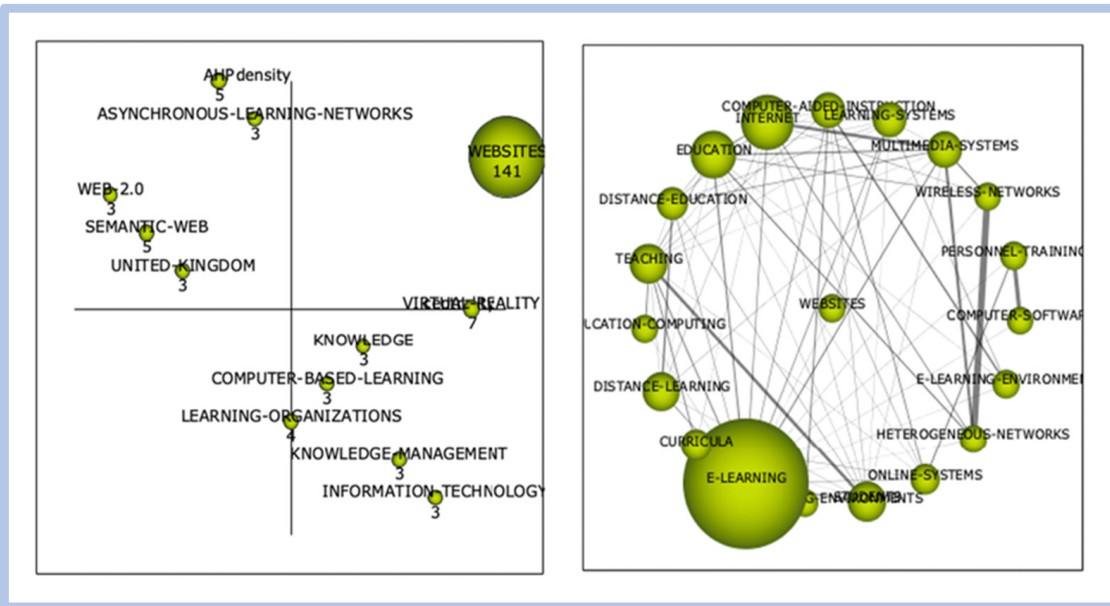

**Figure 10.** E-learning—graphical representation of period 03.

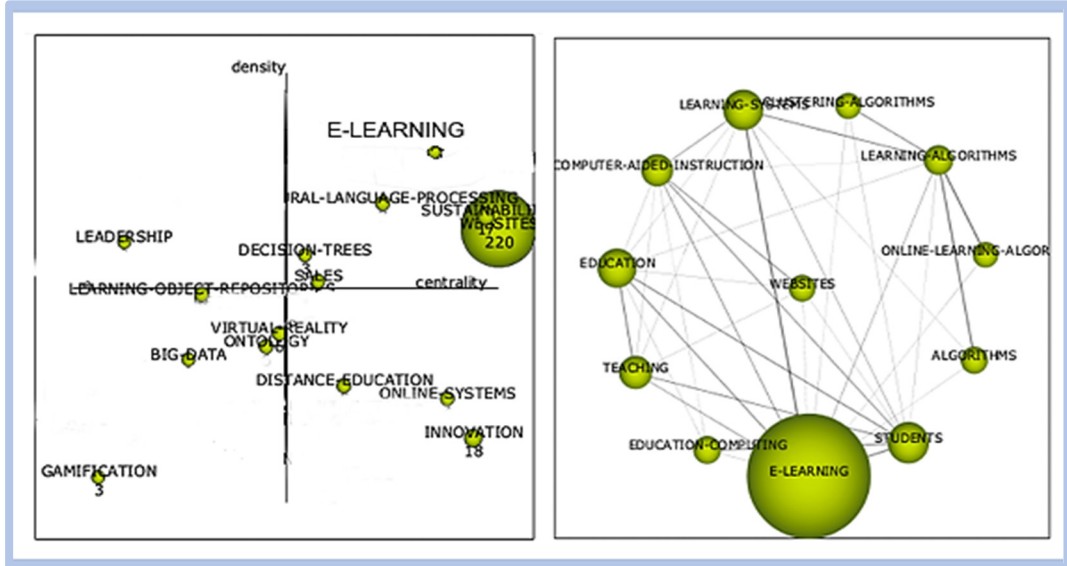

**Figure 11.** Competitiveness and innovation—graphical representation of period 04.

*3.2. Competitiveness and Innovation*

The theme competitiveness innovation has been mentioned since 1970, and had its peak of citations in 2020 with 523 citations. The countries that have published the most on this topic are the United States and China. When analyzing the density and visualization of the network generated from the words competitiveness and innovation, the term innovation has greater relevance over the term competitiveness.

When evaluating the direct relationships of the word innovation, we observed a strong relationship with sustainability, performance, industry 4.0, and strategy. However, there is less of a connection with the words business performance, patents, creativity, and competitive advantage. The strong correlation with the word China (country) is noteworthy, as shown in Figure 12.

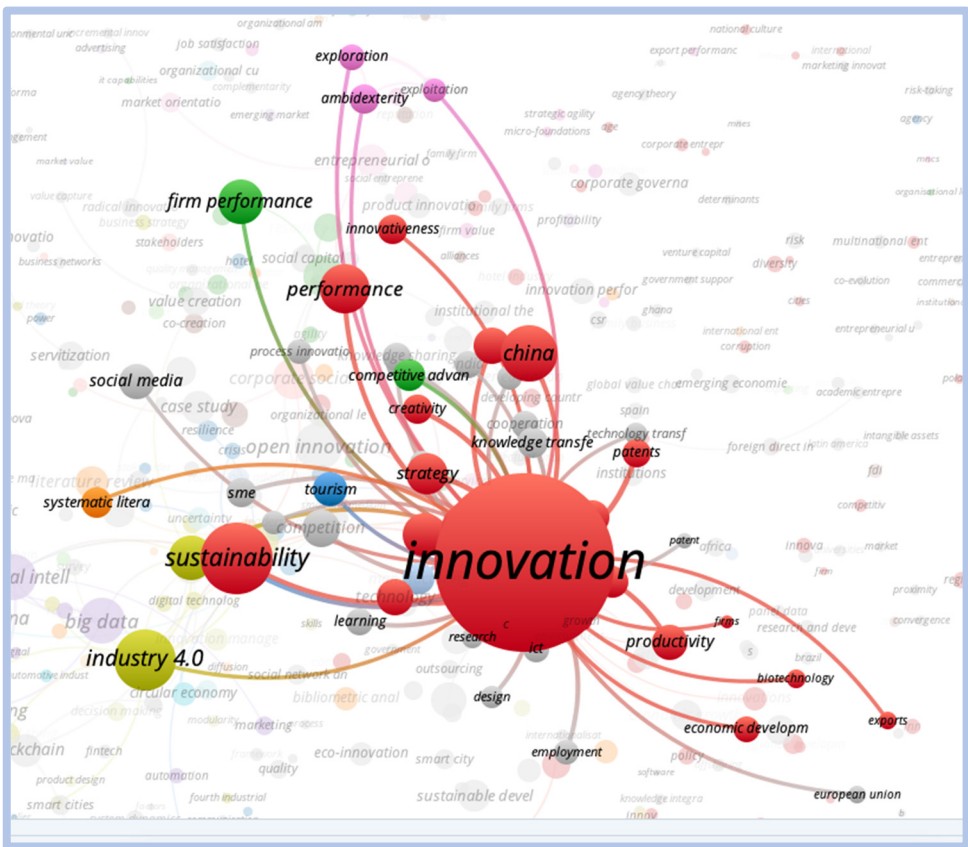

**Figure 12.** Direct relationship—innovation.

There is no direct relationship between the terms "competitiveness and innovation" with the term "e-learning". The closest term to "e-learning" to have a relationship with "competitiveness and innovation" is "knowledge". Regarding the term "critical success factors", no significant correlation was found. The terms closest to the concept of business performance assessment are the terms "business performance" and "performance". These correlations are shown in Figures 13 and 14.

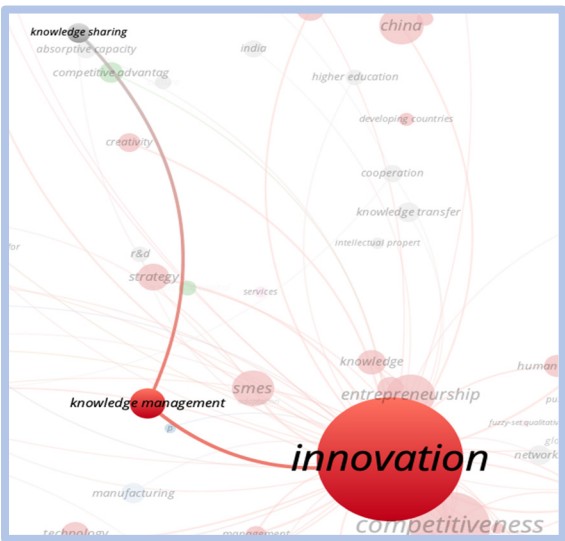

**Figure 13.** Relationship—competitiveness and innovation vs. business performance.

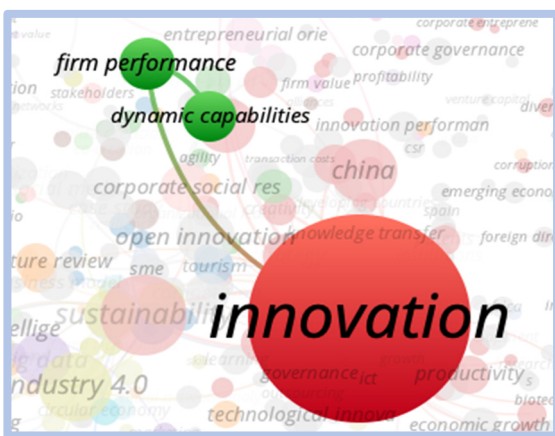

**Figure 14.** Relationship—competitiveness and innovation vs. business performance.

Regarding the term "critical success factors", no significant correlation was found. Using Scimat, we find the evolution of the volume of words used that correspond to the terms "competitiveness and innovation". The evolution of the volume of words associated with the term "competitiveness and innovation" is increasing in the four periods analyzed. The largest volume of words associated with the term occurs between 2011 and 2021, in a total of 18,237 new associations, demonstrating that the volume of research in this area is at its historical peak. Figure 15 shows the evolution of these words over time, divided as follows:

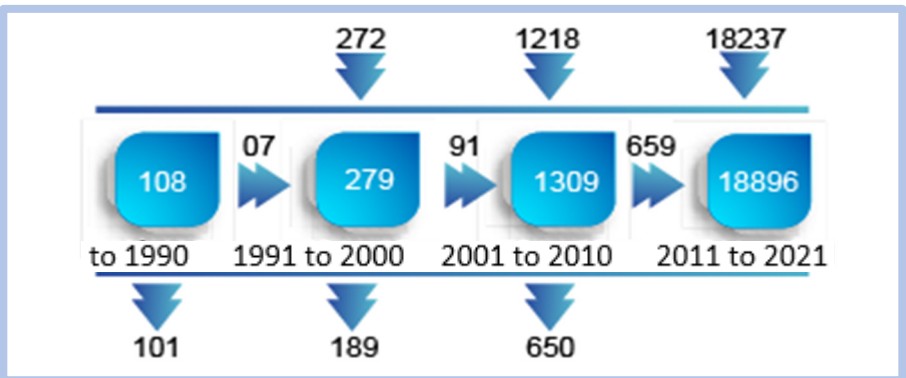

**Figure 15.** Evolution of words over time—competitiveness and innovation.

When the graphic representation of the density and centrality of words in the previously defined periods is verified, we have:

- period 01—until 1990: no graphical representation important for the research;
- period 02—from 1991 to 2000: no graphical representation important for the research;
- period 03—from 2001 to 2010: graphical representation in Figure 16.

In this period, we observe that the term innovation has high centrality and low density, revealing that this cluster has many relationships with other themes. However, these are weak relationships. We also observed that the innovation cluster is directly linked to the competitiveness cluster and this to knowledge, demonstrating a direct relationship between the words and corroborating the application of the term "competitiveness and innovation".

In this period, we observe the displacement of the word innovation from the density of the previous period. When the relationship between word clusters is observed, this relationship becomes clearer with the emergence of new terms such as strategy, knowledge, and creativity, among others.

- period 04—from 2011 to 2021: graphical representation of Figure 17.

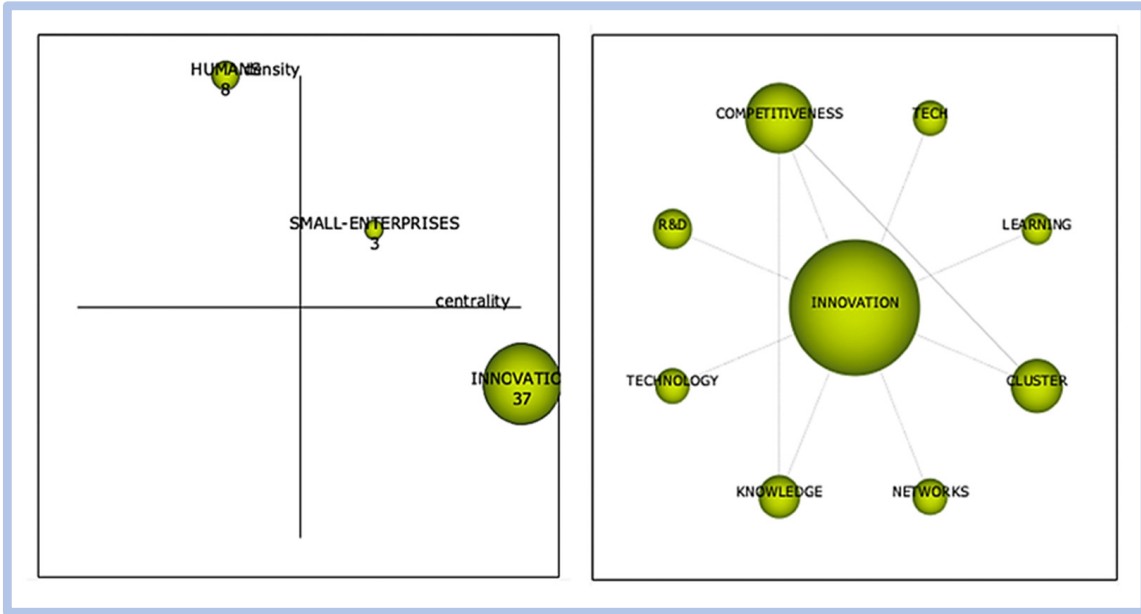

**Figure 16.** Competitiveness and innovation—graphical representation of period 03.

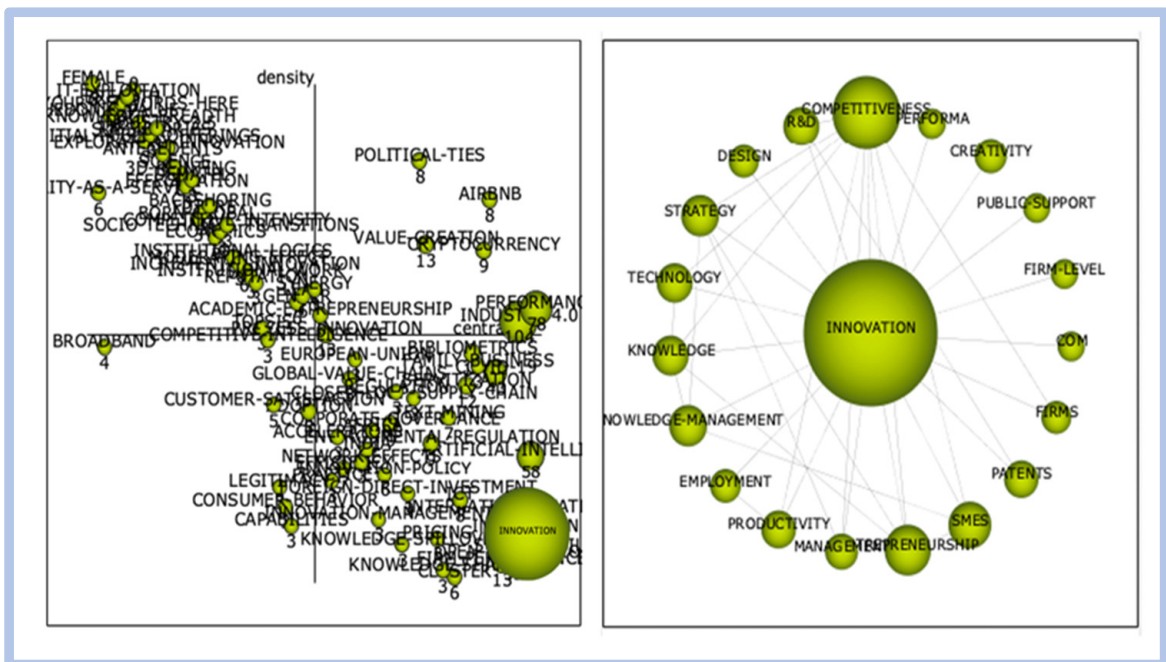

**Figure 17.** Competitiveness and innovation—graphical representation of period 04.

### 3.3. Critical Success Factors

Publications began in mid-1965, peaking in 2019 with 1255 publications that year. The most published author on this topic is Chan, Albert P.C., Hong Kong Polytechnic University, Kowloon—Hong Kong. According to the Scopus database the author was cited in 6834 documents and his "h" index is 57, with a total of 419 publications.

The VOSviewer analysis of the term "critical success factors" draws attention to the clear division of two different clusters. Figure 18 shows the intensity visualization map of these clusters.

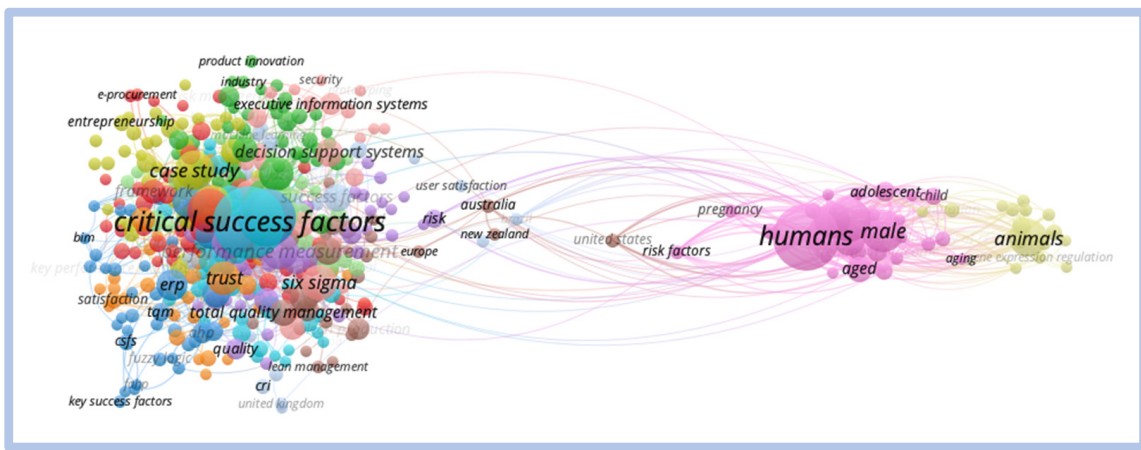

**Figure 18.** Network view—critical success factors.

In the cluster formed by the term "critical success factors" we observe that the terms that are most closely related are case study, implementation, business resource plan, project management, and project success, as shown below in Figure 19:

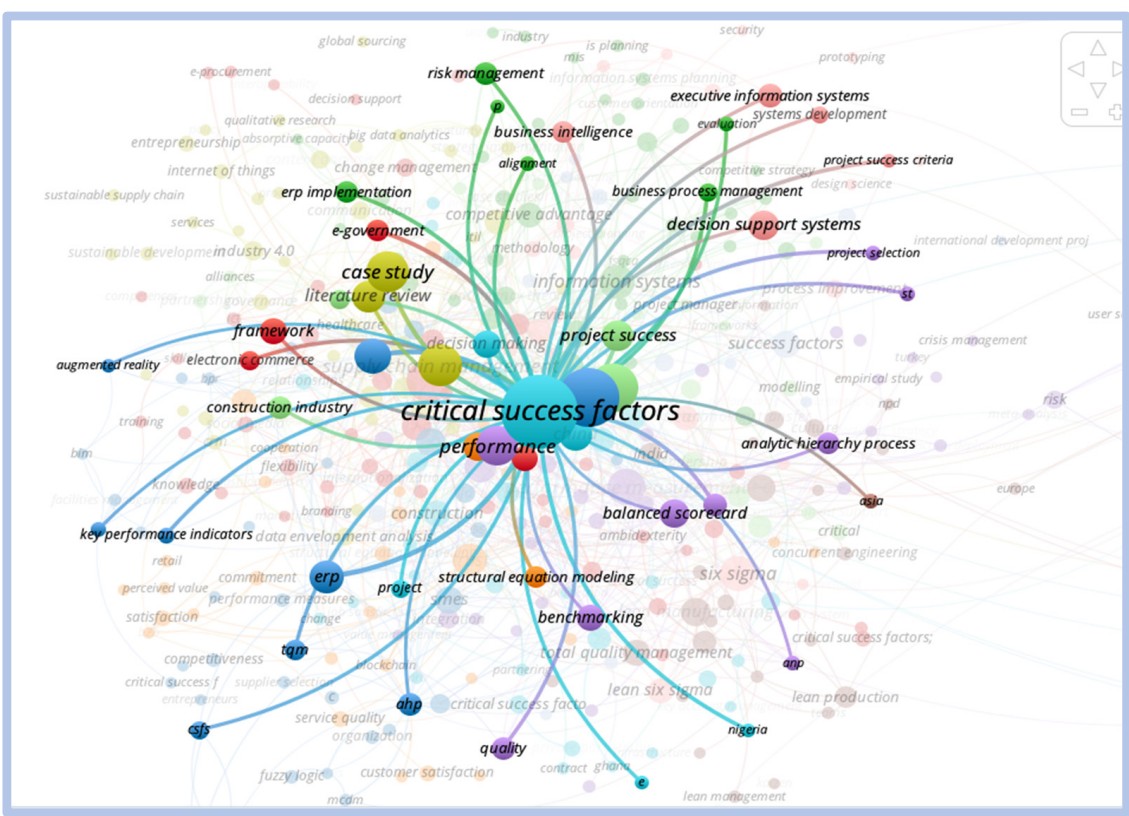

**Figure 19.** Direct relationship—critical success factors.

When we searched for the correlation between the term "critical success factors" and the term "e-learning", we only found a small cluster of the term e-learning isolated. We observe the relationship between the critical success and innovation factors clusters, we observe that this relationship only occurs through the "new products development" cluster, indicating that there is no direct correlation between the themes as shown in Figures 20 and 21 below:

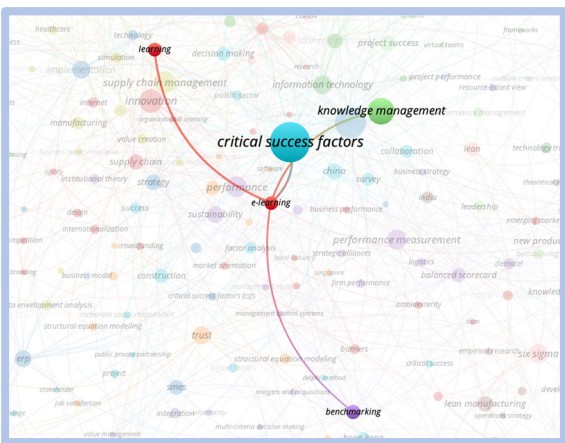

**Figure 20.** Direct relationship—critical success factors vs. e-learning.

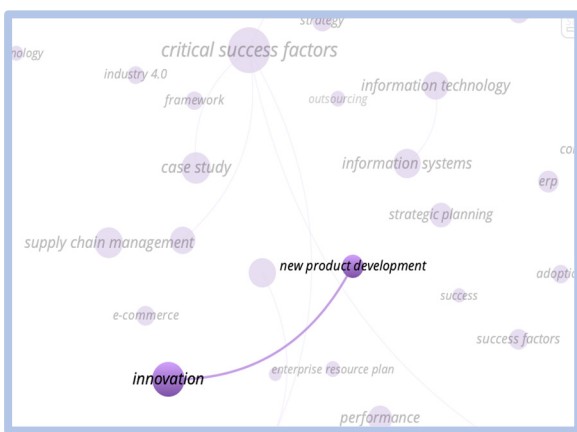

**Figure 21.** Direct relationship—critical success factors vs. e-learning.

With the use of Scimat, we observed an increase in the volume of words associated with the researched topic, showing that the last period has the highest production numbers, as follows in Figure 22.

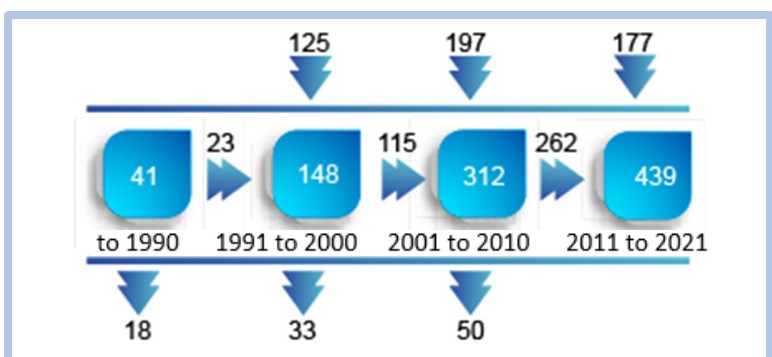

**Figure 22.** Evolution of words over time-critical success factors.

When we observe the evolution of the words, we notice the growth of new associations period by period. As in previous evaluations, the last period has a greater volume, showing a peak of production.

As for the density and centrality of words in the previously defined periods, we have:

- period 01—until 1990: no graphical representation;
- period 02—from 1991 to 2000, as follow in Figure 23.

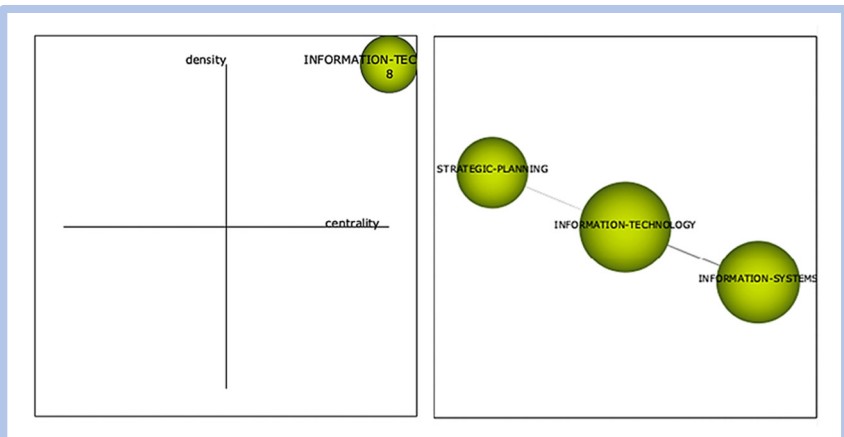

**Figure 23.** Competitiveness and innovation—graphical representation of period 02.

In period 02, the term motor of publications is information technology, and the cluster correlations present in this period for this term are strategic planning and innovation systems. The term critical success factors are not listed in this period.

- period 03—from 2001 to 2010: graphical representation in Figure 24.

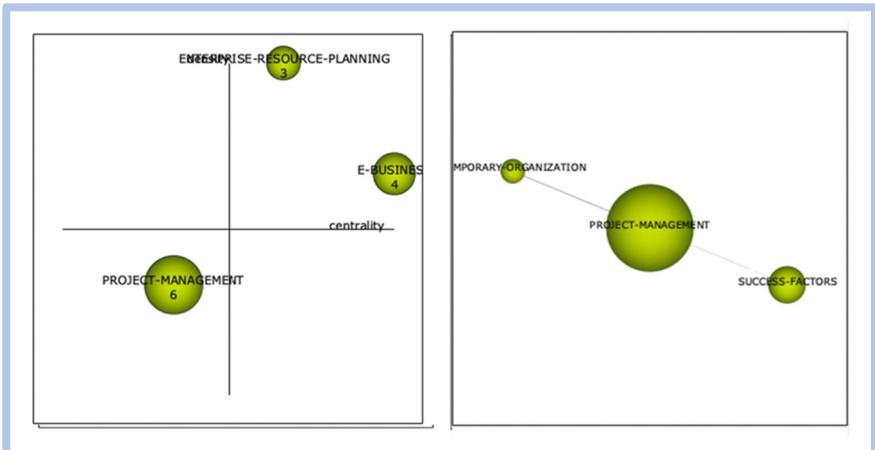

**Figure 24.** Competitiveness and innovation—graphical representation of period 03.

In this period, we observed the terms "e-business" and "enterprise resource planning" as the search engines at the time. The correlation of clusters is from the term "project management" with "success factors" and "temporary organization." In this period, the first more direct reference to the term understudy appears.

In this period, the term critical success factors finally appear in the evaluation with high centrality and low density, revealing that this cluster has many relationships with other themes. However, these relationships were weak. In the correlation between the clusters, "project management" and "case study" are observed as the most relevant interrelated to the cluster under study.

- Figure 25: e-learning—graphical representation of period 04.

To correlate the terms under study, taking into account the methodological importance, it is imperative to pay attention to the adequate scientific basis for a correct approach to the topic [22]. We use the same databases and repeated the same criteria to enter terms. The first term inserted was always e-learning, after which the terms competitiveness and innovation and critical success factors were alternated as filters to know the quantities of files found, thus reducing to the specific term to be evaluated in Table 2:

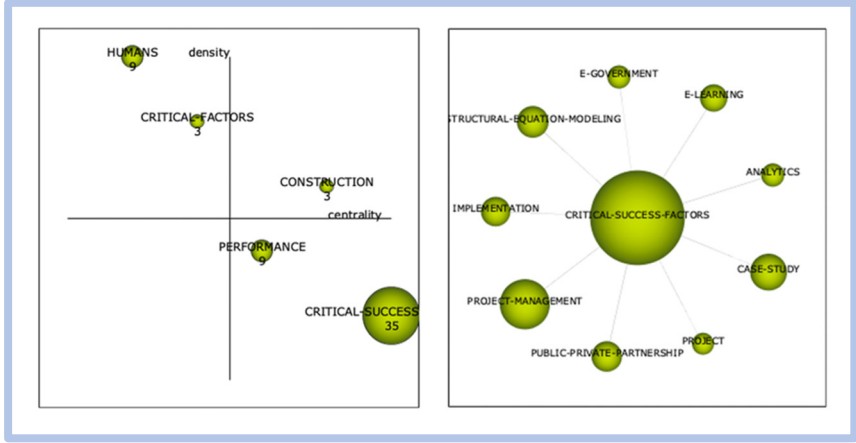

**Figure 25.** Competitiveness and innovation—graphical representation of period 04.

**Table 2.** Publication research configuration parameters on the correlation e-learning × critical success factors × competitiveness and innovation.

| Order of Search Terms:. | Title-Abs-Key: | Number of Articles Founded |
|---|---|---|
| Database: Scopus | | |
| E-learning >> Competitiveness innovation >> critical success factors | ("e-learning") AND (("competitiveness" AND "innovation")) AND ("critical AND success AND factors") AND (LIMIT-TO (DOCTYPE, "ar") OR LIMIT-TO (DOCTYPE, "re")) AND (LIMIT TO (SUBJAREA, "BUSI") OR LIMIT-TO (SUBJAREA, "ECON") OR LIMIT-TO (SUBJAREA, "DECI")) | 12 articles |
| E-learning >> critical success factors >> Competitiveness and innovation | ("e-learning")) AND (("critical AND success AND factors")) AND ("competitiveness" AND "innovation") AND (LIMIT-TO (DOCTYPE, "ar") OR LIMIT-TO (DOCTYPE, "re")) AND (LIMIT-TO (SUBJAREA, "BUSI") OR LIMIT-TO (SUBJAREA, "DECI")) | 9 articles |
| Database: Web of Science | | |
| E-learning >> Competitiveness and innovation >> critical success factors | ("e-learning") Refinado por: TOPIC:("competitiveness" and "innovation") AND CATEGORIES WEB OF SCIENCE: (EDUCATION EDUCATIONAL RESEARCH OR MANAGEMENT OR BUSINESS) AND TYPES OF DOCUMENT: (ARTICLE) Time stipulated: All the years. Índexes: SCI-EXPANDED, SSCI, A and HCI, CPCI-S, CPCI-SSH, ESCI. | 5 articles |
| E-learning >>critical success factors >> Competitiveness and innovation | ("e-learning") Refined by: TÓPIC: ("critical success factors") AND CATEGORIAS DO WEB OF SCIENCE: (EDUCATION EDUCATIONAL RESEARCH OR MANAGEMENT OR BUSINESS) AND TYPES OF DOCUMENT: (ARTICLE OR REVIEW) Time stipulated: All the years. Indexes: SCI-EXPANDED, SSCI, A and HCI, CPCI-S, CPCI-SSH, ESCI. | 41 articles |
| Database: Science Direct | | |
| E-learning >> Competitiveness and innovation >> critical success factors | ("e-learning") AND ("competitiveness" AND "innovation") AND ("critical AND success AND factors") Review articles, Book reviews, Business, Management and Accounting, Decision Sciences | 9 articles |
| E-learning >> critical success factors >> Competitiveness and innovation | ("e-learning") AND ("critical AND success AND factors") AND ("competitiveness" AND "innovation") Review articles, Book reviews, Business, Management, and Accounting | 9 articles |

Source: author.

After identifying the articles in the databases, the repeated articles were deduplicated, resulting in a total of 34 articles. Next, the titles and abstracts were read and those that did not address the subject of this study were excluded. At the end of debugging, 11 articles remained, which are listed in Table 3:

**Table 3.** Articles related to the study topic.

| Article Title | Article | Abstract |
|---|---|---|
| A proposed model of e-learning tools acceptance among university students in developing countries | [30] | Held in Colombia, it proposes an E-Learning Tools Acceptance Model (eLTAM) examine the level of acceptance and critical factors of virtual learning tools among university students in developing countries. |
| Do e-learning service quality influence e-learning student satisfaction and loyalty? Evidence from Vietnam | [31] | Examines the relationships between the attributes of e-learning service quality, overall e-learning service quality, e-learning student satisfaction, and student loyalty in the context of Vietnam. |
| E-Learning Critical Success Factors during the COVID-19 Pandemic: A Comprehensive Analysis of E-Learning Managerial Perspectives | [16] | Seeks to identify the critical success factors for E-learning during COVID-19 using the Multicriteria Analytical Hierarchy Process (AHP) and Technique for Order Preference by Similarity to ideal solution (TOPSIS) to improve the educational process. |
| Emerging themes in e-learning: A review from the stakeholders' perspective | [15] | Reviews emerging issues regarding e-learning methods in organizations. Presents the advantages, disadvantages, challenges, critical success factors, theories, and models from the perspective of stakeholders. |
| Exploring the Factors that Affect Student Satisfaction through Using E-Learning in Malaysian Higher Education Institutions | [32] | A model and instrument were developed to measure student satisfaction with e-learning systems. |
| Incorporating student population differences for effective online education: A content-based review and integrative model | [33] | Reviews and analyzes recent literature relevant to online learning in higher education. Applied a content-based method to collect and evaluate recently published studies related to student populations involved in formal online programs and courses. |
| Online business education research: Systematic analysis and a conceptual model | [34] | Explores authorship, coverage, topicality, context, scope, theories, structures, and key themes through a systematic review of 60 articles from business education journals published since 2008. |
| Prioritizing the components of e-learning systems by using fuzzy DEMATEL and ANP | [35] | Analyzes the relationships of the components of the e-learning systems and prioritizes them in detail for the stakeholders, by using fuzzy DEMATEL. |
| The acceptance of e-learning systems and the learning outcome of students at universities in Vietnam | [36] | Presents research with 357 students from universities in Vietnam. University support, student computer competence, infrastructure, course content and design, and student collaboration have all been influential in the uptake of e-learning. |
| The facilitation of stakeholder consensus for the success of corporate e-learning systems | [17] | Examines enterprise e-learning systems in terms of the role of stakeholders and end-user satisfaction. Identifies potential Critical Success Factors in e-learning systems. Tests whether there is a consensus among stakeholders that these factors facilitate successful implementation. |
| Evaluating critical success factors in implementing E-learning system using multi-criteria decision-making | [20] | Studies the analytical hierarchy process (AHP) with group decision making (GDM) and Fuzzy AHP (FAHP) to study the diverse factors of different dimensions of the web-based E-Learning system. |

Source: author.

## 4. Discussion

E-learning has helped universities to reach students across the planet, reducing costs and investment volume [35], and plays a crucial role in providing educational quality [36]. E-learning is no longer a trend and is a reality [34]. This study sought to understand how e-learning correlates with critical success factors and competitiveness and innovation in the business environment, using research questions and a methodology based on VOSviewer and Scimat software. After the analysis, the response to each question is presented:

### 4.1. Is the Term E-Learning Related to Innovation Research and Business Management?

The co-occurrences of the term e-learning in studies carried out on the terms critical success factors and competitiveness and innovation are incipient. The study found that the volume of research addressing such topics is scarce and that research in this area can grow.

The vision of managing the operation of e-learning as a business has not yet been shown. Administrative approaches in the middle go beyond a more enlightened view of the intrinsic characteristics of the segment. Managing a company that operates in the online education market is considered commonplace, such as any other organization in any other segment.

This understanding is only changed when the market started to become more specialized. That is when competition becomes fiercer when the differentials presented today are just another requirement to be met by all players. Therefore, based on this new scenario, the specialization of management in the e-learning segment tends to be treated with more attention, and studies on the subject are addressed.

A clear example today is that despite this unfavorable scenario, with the advent of the current crisis generated by the COVID-19 pandemic, e-learning research is a growing field. Within a few days, educational organizations that did not work in this segment were forced to adopt electronic teaching methods to keep their students. These institutions had to act so as not to be excluded from the market for not being able to continue delivering knowledge to their students who were in their homes during the lockdown.

Another aspect to be considered, is that in e-learning there are particular business metrics, these are obtained through specific tools. They demand new knowledge about technology, concepts, and understanding of the organizational environment where the company operates. Moreover, in this segment, there are few publications addressing this theme or suggesting methodologies and processes.

### 4.2. What Is the Volume of Scientific Production Covering the Themes E-Learning, Critical Success Factors, and Competitiveness and Innovation?

This study demonstrates that there is not much volume in scientific production that covers the topic of e-learning management. The bibliography, considering the filters applied, had a total of 34 articles after the deduplication of the results obtained from the researched sources. This volume is a possible indicator that there is room for production involving the theme of e-learning in the various aspects of management.

We understand that the volume of publications is on the rise due to the approach that the theme of e-learning is undergoing. The social changes experienced today have brought a new view of teaching methodologies. Moreover, e-learning has undoubtedly had its role re-evaluated and re-signified.

The intellectual production on e-learning management will most likely increase in the coming periods after millions of users who did not even consider its use suddenly had to know, learn, and use e-learning as a means of teaching and learning, due to the lockdown caused by COVID-19.

The volume of publications shows a segment that has not been extensively explored as it is an area that has increased recently in visibility with the growth of technology. Technological evolution brought the modality tools to support the improvement in the quality of education, mainly in developing countries. We can infer that this is because e-learning is researched in its structure, its academic methodology, and the use of technology

that e-learning demands. However, there is little study related to its management, its indicators, and its organizational characteristics. There is much room for new studies to explore the topic.

*4.3. What Is the Trend in E-Learning Research as Related to Managing Critical Success Factors?*

These trends in e-learning research reflect the broader developments in the field and may provide insights into how researchers are exploring the effective management of critical success factors in e-learning initiatives. Some trends are found as follows:

- Personalization and Adaptive Learning: Research in e-learning has focused on developing personalized and adaptive learning approaches. This involves tailoring the learning experience to individual learners' needs, preferences, and progress. Adaptive learning technologies and intelligent tutoring systems are being explored to dynamically adjust the content, pace, and instructional strategies to optimize learner engagement and success.

- Data Analytics and Learning Analytics: E-learning research has increasingly emphasized the use of data analytics and learning analytics to understand and optimize the learning process. By analyzing data collected from learners' interactions with online platforms, researchers can gain insights into learners' behaviors, performance patterns, and engagement levels. This helps in identifying areas for improvement, predicting learner success, and making data-driven decisions to manage critical success factors.

- Social and Collaborative Learning: The importance of social and collaborative learning in e-learning has gained attention. Research has explored the use of social media, online communities, and collaborative tools to foster learner interaction, knowledge sharing, and peer learning. Understanding how to effectively manage and facilitate collaborative learning experiences can contribute to the success of e-learning initiatives.

- Gamification and Immersive Technologies: Gamification elements, such as badges, leaderboards, and rewards, have been studied to enhance learner motivation, engagement, and achievement in e-learning. Additionally, the integration of immersive technologies, such as virtual reality (VR) and augmented reality (AR), has shown potential in creating immersive and interactive learning experiences. Research continues to explore how these approaches can effectively manage critical success factors.

- Mobile Learning and Microlearning: The rise of mobile devices has led to increased research on mobile learning (m-learning) and microlearning. Studies have examined the effectiveness of delivering learning content through mobile devices, optimizing mobile learning experiences, and exploring the potential of microlearning approaches that provide bite-sized, focused learning materials. Understanding how to leverage mobile and microlearning to address critical success factors is an ongoing research area.

- Instructor Support and Training: E-learning research has highlighted the importance of instructor support and training in managing critical success factors. Studies have examined effective strategies for online instructor facilitation, providing timely and constructive feedback, and promoting instructor presence and engagement. Understanding the role of instructors and their impact on e-learning outcomes is crucial for managing success factors.

- Evaluation and Quality Assurance: Research has focused on developing evaluation frameworks and quality assurance mechanisms specific to e-learning. This includes the assessment of course design, content quality, learner satisfaction, and learning outcomes. Understanding how to effectively evaluate and ensure quality in e-learning programs helps in managing critical success factors.

These trends demonstrate that researchers are actively exploring various aspects of e-learning to identify effective strategies for managing critical success factors. By staying updated with the latest research in the field, organizations and e-learning practitioners can gain valuable insights to enhance their approaches and ensure the success of their e-learning initiatives.

## 5. Research Limitations and Future Research Directions

Future research directions reflect the evolving nature of e-learning and highlight areas that researchers may focus on in the future. Some potential future research directions in e-learning:

Artificial Intelligence (AI) in E-Learning: As AI technologies continue to advance, there is scope for further research on integrating AI into e-learning. This includes exploring AI-driven personalized learning, intelligent tutoring systems, natural language processing for automated feedback, and AI-supported assessment and evaluation methods.

Adaptive and Personalized Learning: Future research may delve deeper into the design and implementation of adaptive and personalized learning approaches. This includes investigating algorithms and models for adaptive content delivery, learner modeling techniques, and personalized learning paths that cater to individual learner needs and preferences.

Social and Emotional Aspects of E-Learning: The social and emotional dimensions of e-learning have gained recognition in recent years. Future research may focus on understanding and fostering social presence, collaborative learning, and emotional engagement in online environments. This includes exploring the role of social interactions, peer feedback, and emotional support in promoting effective e-learning experiences.

Ethical Considerations in E-Learning: As e-learning continues to grow, there is a need for research on ethical considerations and challenges. This includes studying issues such as privacy, data security, digital divide, accessibility, and equity in e-learning. Researchers may explore strategies to ensure ethical and inclusive e-learning practices.

Augmented Reality (AR) and Virtual Reality (VR) in E-Learning: AR and VR technologies offer immersive and interactive learning experiences. Future research may focus on exploring the potential of AR and VR in various domains, such as skill training, simulation-based learning, and experiential learning. This includes investigating the effectiveness, usability, and integration of AR and VR into e-learning environments.

Learning Analytics and Educational Data Mining: Research on learning analytics and educational data mining is likely to continue evolving. Future studies may focus on advanced analytics techniques, predictive modeling, and real-time data analysis to gain deeper insights into learner behaviors, performance patterns, and learning outcomes. This includes exploring how learning analytics can inform decision-making and support personalized interventions.

Gamification and Serious Games: Gamification elements and serious games have shown promise in enhancing learner engagement and motivation. Future research may delve into the design and effectiveness of gamified e-learning experiences, the impact of game mechanics on learning outcomes, and the integration of gamification principles into different learning contexts.

Open Educational Resources (OERs) and Open Pedagogy: With the growth of open educational resources, future research may focus on understanding the impact of OER adoption on teaching and learning practices. This includes exploring the effectiveness of open pedagogy approaches, collaborative content creation, and the challenges and benefits of OER integration in e-learning.

Mobile and Ubiquitous Learning: Mobile devices and ubiquitous technologies have transformed the learning landscape. Future research may examine the design and implementation of effective mobile learning experiences, context-aware learning support, and the integration of mobile and ubiquitous technologies into seamless learning environments.

Cross-Cultural and Multilingual E-Learning: With e-learning crossing geographical boundaries, research may explore the challenges and opportunities of cross-cultural and multilingual e-learning. This includes investigating culturally responsive design, strategies for accommodating diverse learners, and effective language support in e-learning platforms.

These potential future research directions highlight the evolving nature of e-learning and the areas that researchers may explore in the coming years. By addressing these

research gaps, the field of e-learning can continue to advance and improve the design, delivery, and impact of online education.

As limitations, we can mention that we only examined the current literature, searching for gaps and opportunities for best practices in e-learning, but it was not possible to compare the literature with the opinion and insights of managers, which also represents a future research opportunity.

## 6. Conclusions

The importance of e-learning, especially in regions that need to spread education to their population, is evident. As for the published studies, we found that this type of teaching does not have a considerable volume of publications, nor does it have managerial approaches consistent with the relevance of the topic. With its dynamic characteristics and appropriation of technologies for teaching distribution, e-learning did not emerge as a driving trend in the studied publications, possibly because the focus of current publications has been on teaching methodologies, tools, and the behavior of students and teachers. Thus, there is an opportunity for future studies to understand how managers can use organizational performance indicators as the critical success factors that can guide management. We also noticed in this study that the publications found are massively addressing the technology aimed at e-learning, the platforms used, and the implementation of e-learning by players in the education segment. There is space to understand how content is delivered locally, the local relationship of e-learning with the community served, and the player's relationship with the locality within the region where it is operating.

**Author Contributions:** Conceptualization, G.A.L., J.C.F. and I.C.B.; methodology, G.A.L., J.C.F. and I.C.B.; software G.A.L., J.C.F. and I.C.B.; validation, G.A.L., J.C.F. and I.C.B..; formal analysis, G.A.L., J.C.F. and I.C.B.; investigation, G.A.L., J.C.F. and I.C.B.; resources, G.A.L., J.C.F. and I.C.B.; data curation, G.A.L., J.C.F. and I.C.B.; writing—original draft preparation, G.A.L., J.C.F. and I.C.B.; writing—review and editing, G.A.L., J.C.F. and I.C.B.; visualization, G.A.L., J.C.F. and I.C.B.; supervision, G.A.L., J.C.F. and I.C.B.; project administration, G.A.L., J.C.F. and I.C.B.; funding acquisition, I.C.B. All authors have read and agreed to the published version of the manuscript.

**Funding:** This research was funded by CAPES—Brazil under the grant CAPES process Baierle No. 88887.464876/2019-00; FAPERGS—Brazil under the grant FAPERGS process Baierle No. 22/2551-0000650-7.

**Informed Consent Statement:** Not applicable.

**Data Availability Statement:** Data will be available on request.

**Conflicts of Interest:** The authors declare no conflict of interest.

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
