# Peer review of "How E-learning Is Correlated with Competitiveness and Innovation and Critical Success Factors"

_education, doi:10.3390/educsci13060619_

Round 1
Reviewer 1 Report
This study employed a systematic literature review to address the e-learning issue. The author/s developed eight steps to transform the data. The findings suggest critical success factors, competitiveness, and innovation. In this paper, the findings are OK, but what are the results associated with previous critical studies? There are some concern points listed as follows:
1. The logic of the judgment is unclear. It did not like the meta-analysis with the well-tested model. At this point, the author/s may need to reinforce the paper's argument with practical meanings.
2. In the structure of the thematic network, each cluster represents the words associated with this theme. This paper argued that the larger the cluster is, the greater the number of associated words. In contrast, the correlation intensity between the clusters is represented by the lines that interconnect them. How has the cluster been done? It is unclear. Should readers believe the figures only? It needs to clarify.
3. The discussion section requires greater external citations/discussion - comparing the results from this study to results obtained in other studies.
4. It would be great if a few examples to elaborate on the findings were cited before building your own premise.
5. The conclusion section needs more work, for example, the practical meanings for educational settings and the suggestion for further studies….
Author Response
Reviewer 1 Comments to Author:
General Comments:
General Comment
This study employed a systematic literature review to address the e-learning issue. The author/s developed eight steps to transform the data. The findings suggest critical success factors, competitiveness, and innovation. In this paper, the findings are OK, but what are the results associated with previous critical studies? There are some concern points listed as follows:
Suggestions:
Suggestion 1 - S#1:
The logic of the judgment is unclear. It did not like the meta-analysis with the well-tested model. At this point, the author/s may need to reinforce the paper's argument with practical meanings
Answer to Suggestion 1 - S#1: Thanks for your comment.
We conducted the meta-analyses using well-established quantitative analysis software in the current literature, such as Scimat and VosViewer. The search results from the databases were input into the software, which, through analysis, cross-referenced the information (keywords, word co-occurrence, etc.) and identified the correlation between the research themes (e-learning and CSF). In addition to correlation, Scimat generates graphs indicating the driving themes, emerging themes, and declining themes. All these meta-analyses are performed by the software and are properly referenced. Due to word limitations, it is not possible to explain the specific commands used by the software, but they are referenced and explained.
Furthermore, to address the comment, we improved the Abstract and section 1.1. In section 2, the steps presented in Figure 2 were further explained. Lastly, sections 4 and 5 were also enhanced and expanded to provide clearer findings based on the meta-analyses.
Suggestion 2 - S#2:
In the structure of the thematic network, each cluster represents the words associated with this theme. This paper argued that the larger the cluster is, the greater the number of associated words. In contrast, the correlation intensity between the clusters is represented by the lines that interconnect them. How has the cluster been done? It is unclear. Should readers believe the figures only? It needs to clarify.
Answer to Suggestion 2 - S#2: Thanks for your suggestion.
We adding text and a new reference to explanation how the cluster are done.
van Eck NJ, Waltman L, Noyons EC, Buter RK. Automatic term identification for bibliometric mapping. Scientometrics. 2010 Mar;82(3):581-596. doi: 10.1007/s11192-010-0173-0. Epub 2010 Feb 11. PMID: 20234767; PMCID: PMC2830586.
Suggestion 3 - S#3:
The discussion section requires greater external citations/discussion - comparing the results from this study to results obtained in other studies.
Answer to Suggestion 3 - S#3: Thanks for your suggestion. To address the comment, we improved the Abstract and section 1.1. Lastly, sections 4 and 5 were also enhanced and expanded to provide clearer findings based on the meta-analyses.
Suggestion 4 - S#4:
It would be great if a few examples to elaborate on the findings were cited before building your own premise
Answer to Suggestion 4 - S#4: Thanks for your suggestion. Sections 4 and 5 were also enhanced and expanded to provide clearer findings based on the meta-analyses.
Suggestion 5 - S#5:
The conclusion section needs more work, for example, the practical meanings for educational settings and the suggestion for further studies.
Answer to Suggestion 5 - S#5: Thanks for your suggestion. Sections 4 and 5 were also enhanced and expanded to provide clearer findings based on the meta-analyses

Reviewer 2 Report
1. The critical research findings related to the title article cannot be seen in the Abstract, and it is suggested that the author(s) should try to rewrite the content of this paragraph based on the final findings.
2. Regarding the keywords critical success factors and their abbreviations CSFs that appear in Abstract, Introduction, and 1.1. E-learning...etc. It is recommended that the author(s) only need to use the full name and add the abbreviation when it appears for the first time, and you can use the abbreviation to express it later.
3. After carefully reading the contents of Introduction and 1.1. E-learning, the reviewer only found that the author(s) gave a good explanation of e-learning. And Innovation, one of the themes of the research, is only partially explained in lines 30-34. For the rest of the focus, such as critical success factors and competitiveness, there is no relevant description or author's definition at all. It is recommended that the author add relevant descriptions in a timely manner so that readers can clearly understand the theme of this study.
4. On page 4, lines 123-124, About Figure 2. Because the research flow will help readers understand the approach of this stage of the study more clearly, it is recommended to enlarge the icon or picture of the research flow and add text descriptions to supplement its completeness.
5. On page 5, line 134, About Figure 3. It is recommended the author(s) confirm whether this figure should be accompanied by the number of relevant literature found in the three database sources mentioned in the study. If so, please add annotations and correct the presentation of this figure.
6. On page 8, line 193. About Figure 6. and 7. Correlation e-learning vs. critical success factors. Please check if there is one item of text in Figure 7 missing.
7. On page 14, line 294. About Figure 20. and 21. Direct relationship - Critical success factors vs e-learning. Please check if there is one item of text in Figure 21 missing.
8. On page 18, lines 389-390. About "As in item 5.1, we understand that the volume of publications is still on the rise due 389 to the approach that the theme of e-learning is undergoing." Please explain what item 5.1 refers to.
9. According to the third research question mentioned on page 2, lines 79 to 80. About "What is the trend in e-learning research as related to managing critical success factors?" The reviewers did not find that the article address the third question in 4. Discussion. It is recommended to add to the findings of the study.
10. It is suggested that a subtitle of research limitations and future research directions can be added before 5. conclusion. This will allow readers to better understand the key points and deficiencies of this research method.
11. On page 19, lines 405-419. Regarding the content of the conclusion, it is suggested that the original intention and findings of this study can be re-examined, and the final discussion corresponding to the title should be adjusted.
12. The text presented in the relevant pictures in this study is a bit too blurry. It is suggested that the text and pictures can be displayed by zooming in.
Finally, the reviewers believe that the advantage of this study is that the relevant process and results are clearly presented and expressed in a step-by-step manner, so that readers can clearly understand the methods and contents of the study. However, the abstract, introduction and conclusion... etc. do not really correspond to the title. It is suggested that the author can re-adjust the content of the relevant chapters according to the above suggestions.
Author Response
Reviewer 2 Comments to Author:
Suggestions:
1. The critical research findings related to the title article cannot be seen in the Abstract, and it is suggested that the author(s) should try to rewrite the content of this paragraph based on the final findings.
Answer to Suggestion 1: Thanks for your comment.
We rewrite the Abstract, adding the following text:
“As a result, it is possible to observe that the co-occurrence of e-learning with critical success factors and competitiveness and innovation is still in its early stages, with scarce research in this area indicating room for future growth. E-learning entails unique business metrics that require specific tools and knowledge of technology, concepts, and the organizational environment. However, there is still a dearth of publications addressing these aspects and proposing relevant methodologies and processes.”
- Regarding the keywords critical success factors and their abbreviations CSFs that appear in Abstract, Introduction, and 1.1. E-learning...etc. It is recommended that the author(s) only need to use the full name and add the abbreviation when it appears for the first time, and you can use the abbreviation to express it later.
Answer to Suggestion 2: Thanks for your comment. We rewrite the text adding the suggestion in Abstract, Introduction and E-learning section.
- After carefully reading the contents of Introduction and 1.1. E-learning, the reviewer only found that the author(s) gave a good explanation of e-learning. And Innovation, one of the themes of the research, is only partially explained in lines 30-34. For the rest of the focus, such as critical success factors and competitiveness, there is no relevant description or author's definition at all. It is recommended that the author add relevant descriptions in a timely manner so that readers can clearly understand the theme of this study.
Answer to Suggestion 3: Thanks for your comment. We rewrite the text adding the following text:
“In the context of e-learning, Critical Success Factors (CSFs) refer to the key factors that are essential for the successful implementation and outcomes of e-learning initiatives or programs. These factors influence the effectiveness, efficiency, and overall success of e-learning efforts. Some common CSFs in e-learning: a) Clear Learning Objectives: Clearly defined and measurable learning objectives are crucial for e-learning success. They provide a clear focus and direction for designing and delivering the online courses or programs; b) Engaging and Interactive Content: E-learning content should be engaging, interactive, and designed to promote active learning. This includes multimedia elements, such as videos, interactive exercises, simulations, and assessments, to enhance learner engagement and knowledge retention; c) User-Friendly Learning Platform: The e-learning platform should be user-friendly, intuitive, and accessible. It should provide a seamless learning experience for learners, allowing them to navigate easily, access resources, participate in discussions, and track their progress; d) Adequate Technical Infrastructure: A reliable and robust technical infrastructure is essential for smooth e-learning delivery. This includes stable internet connectivity, compatible devices, and appropriate software or learning management systems (LMS) that can support the required functionalities; e) Learner Support and Engagement: Providing adequate learner support is crucial in e-learning. This can include access to instructors or mentors, discussion forums, online help resources, and timely feedback to address learners' questions and concerns. Engaging learners through regular communication, feedback, and collaborative activities is also important; f) Effective Assessment and Feedback: A well-designed assessment strategy that aligns with the learning objectives is necessary. It should include various types of assessments, such as quizzes, assignments, and projects, to evaluate learners' understanding and progress. Timely and constructive feedback helps learners improve and stay motivated; g) Pedagogical Design and Instructional Strategies: Applying effective pedagogical principles and instructional strategies in e-learning is critical. This involves designing instructional materials that accommodate diverse learning styles, providing opportunities for active learning and reflection, and incorporating real-world applications; h) Continuous Monitoring and Evaluation: Ongoing monitoring and evaluation of e-learning initiatives are essential to ensure their effectiveness and identify areas for improvement. This includes collecting and analyzing data on learner performance, feedback, and satisfaction, as well as making data-driven decisions to enhance the e-learning experience; i) Scalability and Sustainability: E-learning initiatives should be designed with scalability and sustainability in mind. This involves considering factors such as the potential for growth, the ability to update and adapt content, and the availability of resources to support ongoing e-learning activities.
These CSFs are not exhaustive and may vary depending on the specific context and goals of the e-learning program. It is important for organizations or institutions implementing e-learning to identify and prioritize the critical success factors that align with their unique requirements and desired outcomes.
Competitiveness in the context of e-learning refers to the ability of an e-learning program, platform, or institution to effectively position itself and stand out in the increasingly competitive online education market. It involves offering distinctive features, high-quality content, and a superior learning experience that attracts and retains learners in a highly competitive digital learning environment. Some key aspects of competitiveness in e-learning include: a) Differentiation: To be competitive, e-learning providers need to differentiate themselves from others in the market. This can be achieved through unique course offerings, specialized programs, innovative teaching methods, or by catering to specific target audiences; b) Quality Content and Instruction: Offering high-quality instructional content is essential to remain competitive. This includes well-designed and up-to-date courses, interactive multimedia elements, engaging learning materials, and experienced instructors who can deliver effective online instruction; c) Technological Infrastructure: Competitiveness in e-learning relies on having a robust and user-friendly technological infrastructure. This includes reliable learning management systems (LMS), seamless online platforms, responsive websites or applications, and easy access to course materials; d) Flexibility and Customization: E-learning programs that offer flexibility in terms of scheduling, pacing, and learning pathways are attractive to learners. Customization options, such as personalized learning plans, adaptive learning technologies, and individualized support, can further enhance competitiveness; e) Recognition and Accreditation: Accreditation or recognition by relevant educational bodies or industry associations can enhance the competitiveness of an e-learning program. This adds credibility and demonstrates that the program meets certain quality standards and recognized benchmarks; f) Affordability and Value for Money: Offering competitive pricing structures and value for money is crucial in attracting and retaining learners. Providing transparent pricing, flexible payment options, and demonstrating the return on investment in terms of skills development or career advancement can contribute to competitiveness; g) Continuous Improvement and Innovation: Staying competitive in e-learning requires a commitment to continuous improvement and innovation. This involves monitoring industry trends, adopting emerging technologies, incorporating feedback from learners, and regularly updating course content to remain current and relevant; h) Market Awareness and Marketing Strategies: Understanding the target market, identifying learner needs, and developing effective marketing strategies are essential for competitiveness. This includes market research, targeted advertising, social media presence, and effective communication to attract and engage potential learners.
Competitiveness in e-learning is a dynamic process that requires a proactive approach to adapt to changing market demands, technological advancements, and learner preferences. By focusing on differentiation, quality, engagement, support, flexibility, and continuous improvement, e-learning providers can position themselves competitively and succeed in the online education landscape.”
- On page 4, lines 123-124, About Figure 2. Because the research flow will help readers understand the approach of this stage of the study more clearly, it is recommended to enlarge the icon or picture of the research flow and add text descriptions to supplement its completeness.
Answer to Suggestion 4: Thanks for your comment.
The Figure 2 was recreated according to suggested and reintroduced in the text.
- On page 5, line 134, About Figure 3. It is recommended the author(s) confirm whether this figure should be accompanied by the number of relevant literature found in the three database sources mentioned in the study. If so, please add annotations and correct the presentation of this figure.
Answer to Suggestion 5: Thanks for your comment.
The Figure 3 was recreated according to suggested and reintroduced in the text.
- On page 8, line 193. About Figure 6. and 7. Correlation e-learning vs. critical success factors. Please check if there is one item of text in Figure 7 missing.
Answer to Suggestion 6: Thanks for your comment.
A new figure was added according to the suggested and reintroduced in the text.
- On page 14, line 294. About Figure 20. and 21. Direct relationship - Critical success factors vs e-learning. Please check if there is one item of text in Figure 21 missing.
Answer to Suggestion 7: Thanks for your comment.
A new figure was added according to the suggested and reintroduced in the text.
- On page 18, lines 389-390. About "As in item 5.1, we understand that the volume of publications is still on the rise due 389 to the approach that the theme of e-learning is undergoing." Please explain what item 5.1 refers to.
Answer to Suggestion 8: Thanks for your comment.
The text was corrected according to the suggestion.
- According to the third research question mentioned on page 2, lines 79 to 80. About "What is the trend in e-learning research as related to managing critical success factors?" The reviewers did not find that the article address the third question in 4. Discussion. It is recommended to add to the findings of the study.
Answer to Suggestion 9: Thanks for your comment.
We rewrite the text adding the following text:
“These trends in e-learning research reflect the broader developments in the field and may provide insights into how researchers are exploring the effective management of critical success factors in e-learning initiatives. Some trends are found out as follows:
- Personalization and Adaptive Learning: Research in e-learning has focused on developing personalized and adaptive learning approaches. This involves tailoring the learning experience to individual learners' needs, preferences, and progress. Adaptive learning technologies and intelligent tutoring systems are being explored to dynamically adjust the content, pace, and instructional strategies to optimize learner engagement and success.
- Data Analytics and Learning Analytics: E-learning research has increasingly emphasized the use of data analytics and learning analytics to understand and optimize the learning process. By analyzing data collected from learners' interactions with online platforms, researchers can gain insights into learners' behaviors, performance patterns, and engagement levels. This helps in identifying areas for improvement, predicting learner success, and making data-driven decisions to manage critical success factors.
- Social and Collaborative Learning: The importance of social and collaborative learning in e-learning has gained attention. Research has explored the use of social media, online communities, and collaborative tools to foster learner interaction, knowledge sharing, and peer learning. Understanding how to effectively manage and facilitate collaborative learning experiences can contribute to the success of e-learning initiatives.
- Gamification and Immersive Technologies: Gamification elements, such as badges, leaderboards, and rewards, have been studied to enhance learner motivation, engagement, and achievement in e-learning. Additionally, the integration of immersive technologies, such as virtual reality (VR) and augmented reality (AR), has shown potential in creating immersive and interactive learning experiences. Research continues to explore how these approaches can effectively manage critical success factors.
- Mobile Learning and Microlearning: The rise of mobile devices has led to increased research on mobile learning (m-learning) and microlearning. Studies have examined the effectiveness of delivering learning content through mobile devices, optimizing mobile learning experiences, and exploring the potential of microlearning approaches that provide bite-sized, focused learning materials. Understanding how to leverage mobile and microlearning to address critical success factors is an ongoing research area.
- Instructor Support and Training: E-learning research has highlighted the importance of instructor support and training in managing critical success factors. Studies have examined effective strategies for online instructor facilitation, providing timely and constructive feedback, and promoting instructor presence and engagement. Understanding the role of instructors and their impact on e-learning outcomes is crucial for managing success factors.
- Evaluation and Quality Assurance: Research has focused on developing evaluation frameworks and quality assurance mechanisms specific to e-learning. This includes the assessment of course design, content quality, learner satisfaction, and learning outcomes. Understanding how to effectively evaluate and ensure quality in e-learning programs helps in managing critical success factors.
These trends demonstrate that researchers are actively exploring various aspects of e-learning to identify effective strategies for managing critical success factors. By staying updated with the latest research in the field, organizations and e-learning practitioners can gain valuable insights to enhance their approaches and ensure the success of their e-learning initiatives.”
- It is suggested that a subtitle of research limitations and future research directions can be added before 5. conclusion. This will allow readers to better understand the key points and deficiencies of this research method.
Answer to Suggestion 10: Thanks for your comment.
We rewrite the text adding the following text:
“Future research directions reflect the evolving nature of e-learning and highlight areas that researchers may focus on in the future. Some potential future research directions in e-learning:
Artificial Intelligence (AI) in E-Learning: As AI technologies continue to advance, there is scope for further research on integrating AI into e-learning. This includes exploring AI-driven personalized learning, intelligent tutoring systems, natural language processing for automated feedback, and AI-supported assessment and evaluation methods.
Adaptive and Personalized Learning: Future research may delve deeper into the design and implementation of adaptive and personalized learning approaches. This includes investigating algorithms and models for adaptive content delivery, learner modeling techniques, and personalized learning paths that cater to individual learner needs and preferences.
Social and Emotional Aspects of E-Learning: The social and emotional dimensions of e-learning have gained recognition in recent years. Future research may focus on understanding and fostering social presence, collaborative learning, and emotional engagement in online environments. This includes exploring the role of social interactions, peer feedback, and emotional support in promoting effective e-learning experiences.
Ethical Considerations in E-Learning: As e-learning continues to grow, there is a need for research on ethical considerations and challenges. This includes studying issues such as privacy, data security, digital divide, accessibility, and equity in e-learning. Researchers may explore strategies to ensure ethical and inclusive e-learning practices.
Augmented Reality (AR) and Virtual Reality (VR) in E-Learning: AR and VR technologies offer immersive and interactive learning experiences. Future research may focus on exploring the potential of AR and VR in various domains, such as skill training, simulation-based learning, and experiential learning. This includes investigating the effectiveness, usability, and integration of AR and VR into e-learning environments.
Learning Analytics and Educational Data Mining: Research on learning analytics and educational data mining is likely to continue evolving. Future studies may focus on advanced analytics techniques, predictive modeling, and real-time data analysis to gain deeper insights into learner behaviors, performance patterns, and learning outcomes. This includes exploring how learning analytics can inform decision-making and support personalized interventions.
Gamification and Serious Games: Gamification elements and serious games have shown promise in enhancing learner engagement and motivation. Future research may delve into the design and effectiveness of gamified e-learning experiences, the impact of game mechanics on learning outcomes, and the integration of gamification principles into different learning contexts.
Open Educational Resources (OER) and Open Pedagogy: With the growth of open educational resources, future research may focus on understanding the impact of OER adoption on teaching and learning practices. This includes exploring the effectiveness of open pedagogy approaches, collaborative content creation, and the challenges and benefits of OER integration in e-learning.
Mobile and Ubiquitous Learning: Mobile devices and ubiquitous technologies have transformed the learning landscape. Future research may examine the design and implementation of effective mobile learning experiences, context-aware learning support, and the integration of mobile and ubiquitous technologies into seamless learning environments.
Cross-Cultural and Multilingual E-Learning: With e-learning crossing geographical boundaries, research may explore the challenges and opportunities of cross-cultural and multilingual e-learning. This includes investigating culturally responsive design, strategies for accommodating diverse learners, and effective language support in e-learning platforms.
These potential future research directions highlight the evolving nature of e-learning and the areas that researchers may explore in the coming years. By addressing these research gaps, the field of e-learning can continue to advance and improve the design, delivery, and impact of online education.”
- On page 19, lines 405-419. Regarding the content of the conclusion, it is suggested that the original intention and findings of this study can be re-examined, and the final discussion corresponding to the title should be adjusted.
Answer to Suggestion 11: Thanks for your comment. To address your suggestion, we extended and improved sections 4 and 5 to make the article's contribution and findings clearer. We also modified the title to align it more closely with the conclusions.
- The text presented in the relevant pictures in this study is a bit too blurry. It is suggested that the text and pictures can be displayed by zooming in.
Answer to Suggestion 12: Thanks for your comment.
We recreated the figures according to suggestions and software limitations.
Finally, the reviewers believe that the advantage of this study is that the relevant process and results are clearly presented and expressed in a step-by-step manner, so that readers can clearly understand the methods and contents of the study. However, the abstract, introduction and conclusion... etc. do not really correspond to the title. It is suggested that the author can re-adjust the content of the relevant chapters according to the above suggestions. The logic of the judgment is unclear. It did not like the meta-analysis with the well-tested model. At this point, the author/s may need to reinforce the paper's argument with practical meanings
Answer to final Suggestion: Thanks for your comment.
We rewrite the title with the objective to reflect better the issue of the paper. The new title is: “The relationship of e-learning management with critical success factors and competitiveness and innovation”.

Reviewer 3 Report
Dear authors,
Manuscript is relevant as their conclusions. It is strange for me to look for elearning scientific products without considering the educational framework. It would value the article if you clarify this option.
Author Response
Reviewer 3 Comments to Author:
Suggestions:
1. Manuscript is relevant as their conclusions. It is strange for me to look for elearning scientific products without considering the educational framework. It would value the article if you clarify this option.
Answer to Suggestion 3: Thanks for your comment.
The reason for that option was that the main objective was to study and answer questions related to e-learning management with critical success factors and competitiveness and innovation.

Round 2
Reviewer 2 Report
From the modified content, it is not difficult to find that the author has adjusted it carefully.
Suggestion: Accept in present form.